# Encoding Event-Based Gesture Data With a Hybrid SNN Guided Variational Auto-encoder

## Abstract

Commercial mid-air gesture recognition systems have existed for at least a decade, but they have not become a widespread method of interacting with machines. These systems require rigid, dramatic gestures to be performed for accurate recognition that can be fatiguing and unnatural. To address this limitation, we propose a neuromorphic gesture analysis system which encodes event-based gesture data at high temporal resolution. Our novel approach consists of an event-based guided Variational Autoencoder (VAE) which encodes event-based data sensed by a Dynamic Vision Sensor (DVS) into a latent space representation suitable to compute the similarity of mid-air gesture data. We show that the Hybrid Guided-VAE achieves 87% classification accuracy on the DVSGesture dataset and it can encode the sparse, noisy inputs into an interpretable latent space representation, visualized through T-SNE plots. We also implement the encoder component of the model on neuromorphic hardware and discuss the potential for our algorithm to enable real-time, self-supervised learning of natural mid-air gestures.

## 1 Introduction

Computers that can interact with users through text and spoken language have become ubiquitous. However, people also communicate a great deal of information through body language and mid-air gestures, which are often closely coupled with speech (Fröhlich et al., 2019). This mode of interaction is beneficial in a variety of human computer interaction applications because it is natural, touchless, and efficient. For example, mid-air gestures can be used to interact with car infotainment systems, making interaction safer by eliminating the need for drivers to look at a touch interface instead of the road while driving (Young et al., 2020; May et al., 2017). Mid-air gesture interaction can also be beneficial in high-use public areas, reducing the need for touch based systems which can transmit pathogens from person to person. Additionally, interactive systems that leverage mid-air gestures are more accessible for people with hearing or speech impairments than other touchless alternatives, such as voice (Ballati et al., 2018).

Despite numerous potential applications and recent progress in sensors capable of detecting mid-air gestures (Lichtsteiner et al., 2008b; Zhang, 2012; Keselman et al., 2017; Wang et al., 2016), accurate recognition remains a challenge. Changing backgrounds and lighting conditions, wide variation in how users perform the same gesture, and a need for large data sets are just a few of the reasons that accurate recognition in real-world environments is difficult. Requiring that users perform predefined, rigid gestures is not a solution, as people have difficulty reproducing and remembering rigid, precise movements (Hakim et al., 2019). A reliably accurate gesture recognition system needs to address these issues by recognizing natural gestures despite their wide ranging variability without relying on large data sets recorded in users' homes and sent to the cloud, which would be invasive and a security risk for home owners and their families. One way to address these issues is to adapt in real-time to user's individual gestures by measuring the similarity of incoming gestures to existing gesture classes. On the sensing side, Neuromorphic Dynamic Vision Sensors (DVS) inspired by the biological retina are well suited to this task (Brandli et al., 2014) because they capture temporal, pixel-wise intensity changes as a sparse stream of binary events (Gallego et al., 2019). This approach has key advantages over traditional RGB cameras, such as faster response times, better temporal resolution, and invariance to static image features like lighting and background. Thus, raw DVS sensor data intrinsically emphasizes the dynamic movements that comprise most natural gestures. However, effectively processing DVS event streams remains an open challenge. Events are asynchronous and spatially sparse, making it challenging to directly apply conventional vision algorithms (Gallego et al., 2020; Gallego et al., 2019).

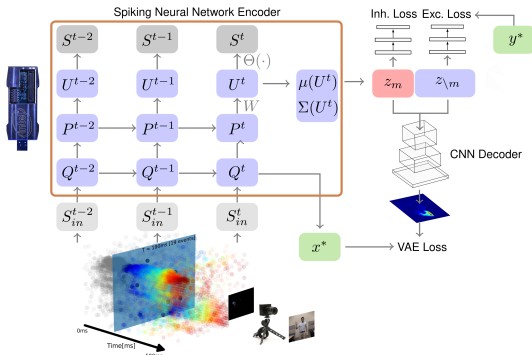

Figure 1: The Hybrid Guided-VAE architecture. Streams of gesture events recorded using a Dynamic Vision Sensor (DVS) are input into a Spiking Neural Network (SNN) that encodes the spatio-temporal features of the input data into a latent structure $z$. $P$ and $Q$ are pre-synaptic traces and $U$ is the membrane potential of the spiking neuron. For clarity, only a single layer of the SNN is shown here and refractory states $R$ are omitted. To help disentangle the latent space, a portion of the $z$ equal to the number of target features $y^*$ is input into a classifier that trains each latent variable to encode these features (Exc. Loss). The remaining $z$, noted $\backslash m$ are input into a different classifier that adversarially trains the latent variables to not encode the target features so they encode for other features instead (Inh. Loss). The latent state $z$ is decoded back into $x^*$ using the conventional deconvolutional decoder layers.

Spiking Neural Networks (SNNs) can efficiently process and learn from event-based data while taking advantage of temporal information (Neftci et al., 2019). SNN models are inspired by the biological cortex and can be used for hierarchical feature extraction from the precise timing of events through event-by-event processing (Gerstner et al., 2014). Recent work demonstrated how SNNs can be trained end-to-end using gradient backpropagation in time and standard autodifferentiation tools, making the integration of SNNs possible as part of modern machine learning and deep learning methods (Zenke & Neftci, 2021; Bellec et al., 2019; Shrestha & Orchard, 2018).

Here, we take advantage of this capability by incorporating a convolutional SNN into a Variational Autoencoder (VAE) to encode spatio-temporal streams of events recorded by the DVS (figure 1). The goal of the VAE is to embed the streams of DVS events into a latent space which facilitates the evaluation of gesture similarity. To best use the underlying hardware, we implement an *hybrid* VAE to process the DVS data, with an SNN-based encoder and a conventional (non-spiking) convolutional network decoder. To ensure the latent space represents features which are perceptually salient and useful for gesture recognition, we use a guided VAE to disentangle the features that account for gesture variation. In the Appendix we show an ablation study that demonstrates the capabilities of the guided VAE to disentangle features of variation over other methods.

Our Hybrid Guided-VAE encodes the gesture data in a way that allows us to automatically measure and analyze the similarity of gestures, allowing us to cluster similar gestures and assign pseudo-labels to novel ones. The key contributions of this work are:

1. End-to-end trainable event-based SNNs for processing neuromorphic sensor data event-by-event and embedding them in a latent space.

2. A Hybrid Guided-VAE that encodes event-based camera data in a latent space representation of salient gesture features for clustering and pseudo-labeling.

3. A proof-of-concept implementation of the Hybrid Guided-VAE on Intel's Loihi Neuromorphic Research Processor.

The ability to encode gestures into a disentangled latent representation from DVS data is a key feature to enable mid-air gesture recognition systems that are less rigid and more natural because they can adapt to each user.

## 2 RELATED WORK

### 2.1 MEASURING GESTURE SIMILARITY

Mid-air gesture data are obtained through gesture elicitation studies, the goal of which is to collect user-defined gestures and then group similar gestures together to determine the gestural language for an application. One way the language is determined is by clustering gestures together based on a set of criteria (Ali et al., 2018) (Villarreal-Narvaez et al., 2020). Examples of criteria used include, but are not limited to, which body part is used, body part trajectory, and body pose. Clustering is often done manually with people assigning labels and providing explanations as to why they labeled a gesture a certain way based on predefined clustering criteria. In this way, gestures frequently used for a particular interaction can be identified. There are several issues with this approach. The clustering criteria used to evaluate which gestures are similar are chosen subjectively by researchers and therefore vary from study to study (Connell et al., 2013a) (Connell et al., 2013b) (Silpasuwanchai & Ren, 2014) (Vatavu, 2012). Using subjective criteria contrasts with the goal of gesture elicitation studies, which is to find an objective consensus among the elicited gestures. The class labels given to the mid air gestures are also subjective and can vary across cultures. After clustering, a similarity score is calculated, with researchers continuously refining the appropriate similarity score to use (Wobbrock et al., 2009) (Vatavu & Wobbrock, 2015) (Tsandilas, 2018) (Vatavu, 2019). Additionally, because human labelers are needed, the process is tedious and expensive, which is why researchers are turning to crowd sourcing to reduce the time taken to cluster gesture data (Ali et al., 2018) (Ali et al., 2019).

To circumvent these issues, our work seeks to automate the process of computing gesture similarity for clustering. With recent advances in deep learning, the process of class labeling for image classification and object recognition tasks can be automated (Bah et al., 2018) or semi-automated (Pugdeethosapol et al., 2020). Because mid air gesture data can be collected in the form of images or video, deep learning can be used for automatic gesture class labeling by learning a similarity measure between the gestures. However, it is still a challenging problem because salient image features may be missing in a scene or can be mistaken as part of the background. Additionally, mid air gestures are dynamic movements, so spatio-temporal information (*i.e.* video) processing is necessary for accurate recognition, which can be computationally expensive. Here, we automate the process of clustering mid air gestures for labeling using spatio-temporal event-based data streamed from a DVS. To automatically and objectively measure the similarity of gestures between and across classes for clustering and automatic labeling, we build a novel Hybrid Guided-VAE model that can take advantage of the DVS' temporal resolution and robustness while being end-to-end trainable using gradient descent on a variational objective. Hybrid VAEs that combine both spiking and ANN layers have been used before on DVS event data for predicting optical flow, with the hybrid architecture efficiently processes the sparse spatio-temporal event inputs while preserving the spatio-temporal nature of the events (Lee et al., 2020).

### 2.2 VARIATIONAL AUTOENCODERS

VAEs are a type of generative model which deal with models of distributions $p(x)$, defined over data points $x \in X$ (Kingma & Welling, 2013). A VAE commonly consists of two networks, 1) an encoder ($Enc$) that encodes the captured dependencies of a data sample $x$ into a latent representation $z$; and 2) a decoder ($Dec$) that decodes the latent representation back to the data space making a reconstruction $\tilde{x}$. Using Gaussian assumptions for the latent space :

$$Enc(x) = q(z|x) = N(z|\mu(x), \Sigma(x)), \tag{1}$$

$$\tilde{x} \approx Dec(z) = p(x|z), \tag{2}$$

where $q$ is the encoding probability model into latent states $z$ that are likely to produce $x$, and $p$ is the decoding probability model conditioned $z$. The functions $\mu(x)$ and $\Sigma(x)$ are deterministic functions whose parameters can be trained through gradient-based optimization. Using a variational approach, the VAE loss consists of the sum of two terms resulting from the variational lower bound:

$$\log p(x) \geq \underbrace{\mathbb{E}_{z \sim q} \log p(x|z)}_{\mathcal{L}_{ll}} - \underbrace{D_{KL}(q(z|x)||p(z))}_{\mathcal{L}_{prior}}. \tag{3}$$

The first term is the expected log likelihood of the reconstructed data computed using samples of the latent space, and the second term acts as a prior, where $D_{KL}$ is the Kullback-Leibler divergence. The *VAE* loss is thus formulated to maximize the variational lower bound by maximizing $-\mathcal{L}_{prior}$ and $\mathcal{L}_{ll}$. VAE's latent space captures salient information for representing the data $X$, and thus similarities in the data (Larsen et al., 2016).

## 2.3 DISENTANGLING VARIATIONAL AUTOENCODERS

VAEs do not necessarily disentangle all the factors of variation, which can make the latent space difficult to interpret and use. Several approaches have been developed to improve the disentangling of the latent representation, such as Beta VAE (Higgins et al., 2017) and Total Correlation VAE (Chen et al., 2018). For this reason, we employ a Guided-VAE, which has been developed specifically to disentangle the latent space representation of key features in a supervised fashion (Ding et al., 2020). We describe here the supervised Guided-VAE algorithm, which is the basis of our hybrid model described in the next section.

To learn a disentangled representation, a supervised Guided-VAE trains latent variables to encode existing ground-truth labels while making the rest of the latent variables uncorrelated with that label. The supervised Guided-VAE model targets the generic generative modeling task by using an adversarial excitation and inhibition formulation. This is achieved by minimizing the discriminative loss for the desired latent variable while maximizing the minimal classification error for the rest of the latent variables. For $N$ training data samples $X = (x_1, ..., x_N)$ and $M$ features with ground-truth labels, let $z = (z_1, ..., z_m, ...z_M) \oplus z_{\backslash m}$ where the $z_m$ define the "guided" latent variable capturing feature $m$, and $z_{\backslash m}$ represents the rest of the latent variables. Let $y_m(x_n)$ be a one-hot vector representing the ground-truth label for the $m$-th feature of sample $x_n$. For each feature $m$, the excitation and inhibition losses are defined as follows:

$$
\begin{aligned}
\mathcal{L}_{Exc}(z, m) &= \max_{c_m} \left( \sum_{n=1}^{N} \mathbb{E}_{q(z_m|x_n)} \log p_{c_m}(y = y_m(x_n)|z_m) \right), \\
\mathcal{L}_{Inh}(z, m) &= \max_{k_m} \left( \sum_{n=1}^{N} \mathbb{E}_{q(z_{\backslash m}|x_n)} \log p_{k_m}(y = y_m(x_n)|z_{\backslash m}), \right)
\end{aligned}
\tag{4}
$$

where $c_m$ is a classifier making a prediction on the $m$-th feature in the guided space and $k_m$ is a classifier making a prediction over $m$ in the unguided space $z_{\backslash m}$. By training these classifiers adversarially with the VAE's encoder, it learns to disentangle the latent representation, with $z_m$ representing the target features and $z_{\backslash m}$ representing any features other than the target features.

## 3 METHODS

### 3.1 DYNAMIC VISION SENSORS AND PREPROCESSING

Dynamic Vision Sensors (DVS) are a type of event-based sensor that record event streams at a high temporal resolution and are compatible with SNNs (Liu & Delbruck, 2010). Inspired by the human retina, DVS sensors detect brightness changes on a logarithmic scale with a user-tunable threshold, instead of RGB pixels like typical cameras. An event consists of its location $x$, $y$, timestamp $t$, and the polarity $p \in [\text{OFF}, \text{ON}]$ representing the direction of change. Using a dense representation, the DVS event stream is denoted $S^t_{DVS,x,y,p} \in \mathbb{N}^+$, indicating the number of events that occurred in the time bin $(t, t + \Delta t)$ with space-time coordinates $(x, y, p, t)$. $\Delta t$ is the temporal discretization and equal to 1ms in our experiments. The recorded DVS event stream is provided as input to the Hybrid Guided-VAE network implemented on a GPU (network dynamics described in the following section). Each polarity of the event stream $(x, y, p)$ is fed onto one of the two channels of the first convolutional layer. Within the time step, most pixels have value zero, and very few have values larger than two. Note that time is *not* represented as a separate dimension in the convolutional layer, but through the dynamics of the SNN.

The VAE targets take a different form because the decoder is not an SNN. Time surfaces (TS) are widely used to preprocess event-based spatio-temporal features (Lagorce et al., 2017). TS can be constructed by convolving an exponential decay kernel through time in the event stream as follows:

$$
TS^t_{x,y,p} = \epsilon^t * S^t_{DVS,x,y,p} \text{ with } \epsilon^t = e^{-\frac{t}{\tau}}
\tag{5}
$$

where $\tau$ is a time constant. Here, we convolve over the time length of the input gesture data stream $S^t_{DVS,x,y,p}$. This results in two 2D images, one for each polarity, that are used as VAE targets( *i.e.*, for the reconstruction loss).

## 3.2 Hybrid Guided Variational Auto-Encoder

Gestures recorded using a DVS camera produce streams of events containing rich spatio-temporal patterns of the gesture. Event-based computer vision algorithms typically extract hand encoded statistics and use these in their models. While efficient, this approach discards important spatio-temporal features from the data (Gallego et al., 2019).

Rather than manually selecting a feature set, we process the raw DVS events while preserving key spatio-temporal features using a spiking neural network (SNN) trained end-to-end in the Hybrid Guided-VAE architecture shown in Figure 1.

A key advantage of the VAE is that the loss can be optimized using gradient backpropagation. To retain this advantage in our hybrid VAE, we must ensure that the encoder SNN is also trainable through gradient descent. Until recently, several challenges hindered this: the spiking nature of neurons' nonlinearity makes it non-differentiable and the continuous-time dynamics raise a challenging temporal credit assignment problem. These challenges are solved by the surrogate gradients approach (Neftci et al., 2019), which formulates the SNN as an equivalent binary RNN, and employs a smooth surrogate network for the purposes of computing the gradients. Our Hybrid Guided-VAE uses a convolutional SNN to encode the spatio-temporal streams in the latent space, and a non-spiking convolutional decoder to reconstruct the TS of the data. We chose an event-based encoder because the SNN can bridge computational time scales by extracting slow and relevant factors of variation in the gesture (Wiskott & Sejnowski, 2002) from fast event streams recorded by the DVS. We chose a conventional (non-spiking) decoder for three reasons: (1) for gesture similarity estimation, we are mainly interested in the latent structure produced by the encoder, rather than the generative features of the network, (2) as we demonstrate in the results section, a dedicated neuromorphic processor (Indiveri et al., 2011; Davies et al., 2018) only requires the encoder to produce this latent structure, and (3) SNN training is compute- and memory- intensive. Thus a conventional decoder enables us to dedicate more resources to the SNN encoder.

Our Hybrid Guided-VAE network architecture is shown in Figure 1 and the architecture description is provided in Table 1. Descriptions of the excitatory and inhibitory networks necessary for "guiding" of the VAE are provided in the Appendix. The SNN encoder consists of four discrete leaky integrate and fire convolutional layers (see following section) followed by linear layers, and outputs a pair of vectors $(\mu, \Sigma)$ for sampling the latent state $z$. According to the guided VAE, part of or all of latent state $z$ is inputed into one of three connected networks, the excitation classifier, the adversarial inhibition classifier, or the decoder. The target features for the excitatory network are given as one-hot encoded vectors of length $M$. The excitation classifier is jointly trained with the encoder to train the first $M$ latent variables to only encode information relevant to the corresponding target feature. The inhibition classifier takes as input the remaining latent variables in the latent space, $z_{\backslash m}$, and are adversarially trained on two sets of targets. One set of targets are the same target features that the excitation classifier trains on. The other set of target features is a vector of length $M$ but all of the values are set to $0.5$ indicating that none of the values correspond to any target. The inhibition classifier is jointly trained with the encoder to train the remaining $z_{\backslash m}$ latent variables to not encode any information relevant to target features forcing them to instead encode information for other features. The decoder is a transposed convolutional network that takes the full latent state $z$ as input to construct the TS, denoted $\tilde{x}$ in Figure 1.

Table 1: Hybrid Guided-VAE architecture

| Layer | Kernel | Output | Layer Type |
|---|---|---|---|
| input | | $32{\times}32{\times}2$ | DVS128 |
| 1 | 2a | $16{\times}16{\times}2$ | SNN LIF Encoder |
| 2 | 32c7p0s1 | $16{\times}16{\times}32$ | |
| 3 | 1a | $16{\times}16{\times}32$ | |
| 4 | 64c7p0s1 | $16{\times}16{\times}64$ | |
| 5 | 2a | $8{\times}8{\times}64$ | |
| 6 | 64c7p0s1 | $8{\times}8{\times}64$ | |
| 7 | 1a | $8{\times}8{\times}64$ | |
| 8 | 128c7p0s1 | $8{\times}8{\times}128$ | |
| 9 | 1a | $8{\times}8{\times}128$ | |
| 10 | - | $128$ | |
| 11 | - | $100$ | $\mu(U^t)$ (ANN) |
| 12 | - | $100$ | $\Sigma(U^t)$ (ANN) |
| 13 | - | $128$ | ANN Decoder |
| 14 | 128c4p0s2 | $4{\times}4{\times}128$ | |
| 15 | 64c4p1s2 | $8{\times}8{\times}64$ | |
| 16 | 32c4p1s2 | $16{\times}16{\times}32$ | |
| output | 2c4p1s2 | $32{\times}32{\times}2$ | Time Surface |

Notation: `Ya` represents `YxY` sum pooling, `XcYpZsS` represents `X` convolution filters (`YxY`) with padding $Z$ and stride $S$.

### 3.3 ENCODER SNN DYNAMICS

To take full advantage of the event-based nature of the DVS input stream and its rich temporal features, the data is encoded using an SNN. SNNs can be formulated as a type of recurrent neural network with binary activation functions (Figure 1) (Neftci et al., 2019). With this formulation, SNN training can be carried out using standard tools of autodifferentiation. In particular, to best match the dynamics of existing digital neuromorphic hardware implementing SNNs (Davies et al., 2018; Furber et al., 2014), our neuron model consists of a discretized Leaky Integrate and Fire (LIF) neuron model with time step $\Delta t$ (Kaiser et al., 2020):

$$U_i^t = \sum_j W_{ij} P_j^t - U_{th} R_i^t + b_i, \qquad P_j^{t+\Delta t} = \alpha P_j^t + (1-\alpha) Q_j^t,$$
$$S_i^t = \Theta(U_i^t), \qquad\qquad Q_j^{t+\Delta t} = \beta Q_j^t + (1-\beta) S_{in,j}^t, \tag{6}$$

where the constants $\alpha = \exp(-\frac{\Delta t}{\tau_{\mathrm{mem}}})$ and $\beta = \exp(-\frac{\Delta t}{\tau_{\mathrm{syn}}})$ reflect the decay dynamics of the membrane potential and the synaptic state during a $\Delta t$ timestep, where $\tau_{mem}$ and $\tau_{syn}$ are membrane and synaptic time constants, respectively. The time step in our experiments was fixed to $\Delta t = 1$ms. $R_i$ here implements the reset and refractory period of the neuron (with dynamcis similar to $P$), and states $P_i$, $Q_i$ are pre-synaptic traces that capture the leaky dynamics of the membrane potential and the synaptic currents. $S_i^t = \Theta(U_i^t)$ represents the spiking non-linearity, computed using the unit step function, where $\Theta(U_i) = 0$ if $U_i < U_{th}$, otherwise 1. We distinguish here the input spike train $S_{in}^t$ from the output spike train $S^t$. Following the surrogate gradient approach (Neftci et al., 2019), for the purposes of computing the gradient, the derivative of $\Theta$ is replaced with the derivative of the fast sigmoid function (Zenke & Ganguli, 2017). Note that equation (6) is equivalent to a discrete-time version of the spike response model with linear filters (Gerstner & Kistler, 2002). Similar networks were used for classification tasks on the DVSGesture dataset, leading to state-of-the-art accuracy on that task (Kaiser et al., 2020; Shrestha & Orchard, 2018). In the Appendix, we show how these equations can be obtained via discretization of the common LIF neuron.

The SNN follows a convolutional architecture, as described in Table 1, encoding the input sequence $S_{in}^t$ into a membrane potential variable $U^t$ in the final layer. The network computes $\mu(U^t)$ and $\Sigma(U^t)$ as in a conventional VAE, but uses the final membrane potential state $U^t$. Thanks to the chosen neural dynamics, the TS can be naturally computed by our network. In fact, using an appropriate choice of $\tau = \tau_{syn}$ for computing the TS, it becomes exactly equivalent to the pre-synaptic trace $Q^t$ of our network (See Appendix). Hence, our choice of input and target corresponds to an autoencoder in the space of pre-synaptic traces $Q^t$.

### 3.4 DATASETS

We trained and evaluated the model using the Neuromorphic MNIST (NMNIST) and IBM DVS-Gesture datasets, both of which were collected using DVS sensors (Lichtsteiner et al., 2008a; Posch et al., 2011). NMNIST consists of $32 \times 32$, 300ms event data streams of MNIST images recorded with a DVS (Orchard et al., 2015). The dataset contains 60,000 training event streams and 10,000 test event streams.

The IBM DVSGesture dataset (Amir et al., 2017) consists of recordings of 29 different individuals performing 10 different gestures, such as clapping, and an 'other' gesture class containing gestures that do not fit into the first 10 classes (Amir et al., 2017). The gestures are recorded under four different lighting conditions, so each gesture is also labeled with the associated lighting condition under which it was performed. Detailed analyses of the 'other' class and the four lighting conditions are included in the Appendix. Samples from the first 23 subjects were used for training and the last 6 subjects were used for testing. The training set contains 1078 samples and the test set contains 264 samples. Each sample consists of about 6 seconds of the gesture being repeatedly performed. In our work we scale each sample to 32x32 and only use a randomly sampled 200ms sequence to match real time learning conditions. For both datasets, to reduce memory requirements, the gradients were truncated to 100 time steps (*i.e.* 100ms worth of data).

For both datasets, the model learns a latent space encoding that can be used to reconstruct the digits or gestures and to classify instances accurately.

## 3.5 Neuromorphic Hardware Implementation

As a first step towards a self-supervised gesture recognition system based on the Hybrid Guided-VAE, we developed a proof-of-concept implementation of our SNN encoder which can be trained and run on Intel Loihi. Since only the encoder is required for estimating the feature embeddings, it is not necessary to map the decoder onto the Loihi. We trained the neuromorphic encoder with our Hybrid Guided-VAE method with a couple key differences in order for the encoder to work on Loihi as follows: 1) To train with the same neuron model and quantization as the Loihi we used a differentiable functional simulator of the Loihi chip (Shrestha & Orchard (2018); Stewart et al. (2020a)), which allows one-to-one mapping of trained networks onto the hardware; and 2) The $\mu$ and $\Sigma$ parts of the network were made spiking and used the quantized membrane potential of the neuron for the latent representation instead of ANN trained full precision values.

## 4 Results

### 4.1 NMNIST Accuracy and Latent Space

The use of the NMNIST spiking digit classification data allows us to test the ability of the algorithm to learn to encode spiking input data in a manner that preserves information needed for accurate classification and captures salient features in the latent space. Trained on the NMNIST dataset, the excitation classifier achieved both training and test accuracy of approximately 99% indicating that the SNN encoder learned a latent representation that clearly disentangles digit classes.

We used T-SNE to visualize the learned representations of the algorithm. T-SNE embeds both the local and global topology of the latent space into a low-dimensional space suitable for visualization (Mukherjee et al., 2019), allowing us to observe clustering and separation between gesture classes and lighting conditions. As shown in Figure 2, each digit class in the NMNIST dataset is clearly separable in the latent space, with only a few data points inaccurately clustered.

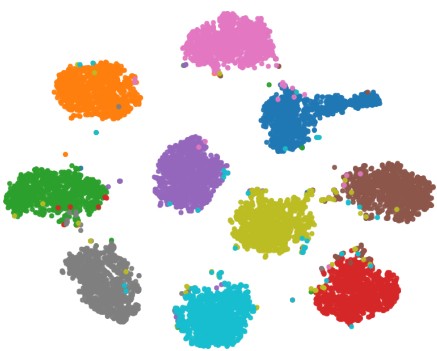

Figure 2: A T-SNE plot of the $z_m$ portion of the latent space of the encoded NMNIST dataset. Color of the data points corresponds to the digit classes. Clear separation between classes indicates that the algorithm learns to encode the spiking data into a latent space that strongly emphasizes class-relevant features over other variation.

### 4.2 DVSGesture Accuracy, Latent Space, and Reconstructions

To analyze the learned representation of gestures in the latent space we examine the accuracy of the excitation classifier in correctly identifying a gesture, the T-SNE projections of the different parts of the latent space, and traversals of the latent space. We compared our results against partially ablated models (no-guiding, and no-spiking convolutional encoder) with results in the Appendix. Finally, we observe the quality of the embeddings based on our own DVS recordings of novel gestures.

The excitation classifier results on the DVSGesture dataset achieves a training accuracy of approximately 97% and test accuracy of approximately 87%. Qualitatively, the SNN encoder learns a disentangled latent representation of features unique to each gesture class but has some difficulty distinguishing between gestures that are very similar.

Figure 3: Original (top) and reconstructed (bottom) time-surfaces for a sample gesture from each class. The reconstructions reflect the location of each gesture but with some smoothing of the detail.

In Figure 3, a sample gesture from each of the gesture classes is visualized as a TS. Colors in the samples correspond to the TS value of the events at the end of the sequence. Note that the TS leaves a significant amount of fine detail intact. In contrast, encoding and then decoding the samples results in a reconstruction that preserves the general structure of the gesture but smooths out some of the detail. Note that, for the purposes of estimating gesture similarity, the fidelity of the reconstruction is less important than the capacity of the model to disentangle the gesture classes. Furthermore, disentangling autoencoders are known to provide lower quality reconstructions compared to unguided VAEs (Chen et al., 2018).

We use T-SNE to examine the capacity of the network to disentangle salient gesture features in the latent space. Figure 4 (Left) shows a T-SNE plot of the guided portion of the latent space trained on the DVSGesture dataset. This plot indicates clear clustering of gesture classes, with some overlap between similar gestures such as left arm clockwise and counterclockwise. This global structure in the learned representations indicates the Hybrid Guided-VAE is identifying useful, class-relevant features in the encoder and suppressing noise and unhelpful variability from the spiking sensor.

To test the generalization of the learned encoder, we evaluated how the VAE model performs when provided new gesture data captured in a new environment intended to replicate ecologically valid conditions for a real-world gesture recognition system. We recorded gestures belonging to two new classes, right and left swipe down, which are not present in the DVSGesture dataset. We used a different DVS sensor (the DAVIS 240C sensor (Brandli et al., 2014)) and processed the data with the trained Hybrid Guided-VAE. Each gesture was repeated three times for approximately 3s by the same subject under the same lighting conditions.

Figure 4 (Left) shows the new gesture TS and associated T-SNE embeddings in the $z_m$ portion of the latent space. The right swipe down gestures were represented by the model similar to right hand wave gestures, as indicated by their proximity in the T-SNE plot of the latent space. Similarly, the left swipe down gestures were represented most similar to left hand wave gestures. Interestingly, both new classes cluster near the edges of the existing classes, possibly indicating the presence of a

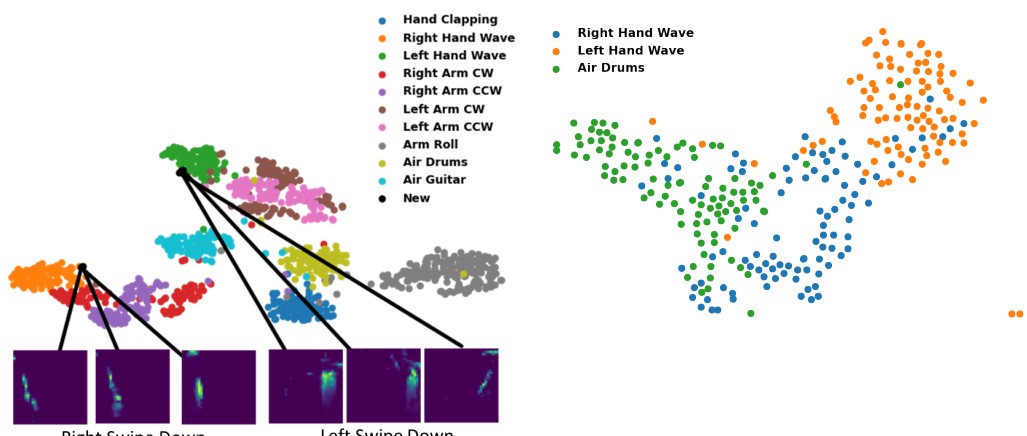

Figure 4: (Left) A T-SNE plot of the $z_m$ portion of the latent space of the encoded DVSGesture dataset. Additionally, projections of the $z_m$ portion of the latent space of encoded new gestures we recorded using a different DVS, and not part of the DVSGesture dataset are shown. Bottom color plots are the TS of the new gestures. (Right) T-SNE plot of the $z_m$ portion of the latent space disentanglement of three gesture classes implemented on the Intel Loihi.

feature gradient. These results demonstrate that the Hybrid Guided-VAE is capable of appropriately representing novel gestures in a manner that could support pseudolabeling. With additional data points of new gestures, the hybrid Guided-VAE can eventually learn new classes of gestures on it own.

As an additional tool to investigate the structure of the latent space representations, we use latent traversals to interpret the features of the space. Traversals consist of a set of generated TS based on positions in the space computed using off polarity events. The positions traverse a line in the latent space, revealing how the network encodes salient features.

Figure 5 contains two traversals illustrating the shift in position and intensity of motion in the TS encoded by the latent variables $z_2$ and $z_3$. Because those features correspond to distinguishing, salient characteristics of the gesture classes "Right Hand Wave" and "Left Hand Wave", this encoding allows the model to disentangle the gestures.

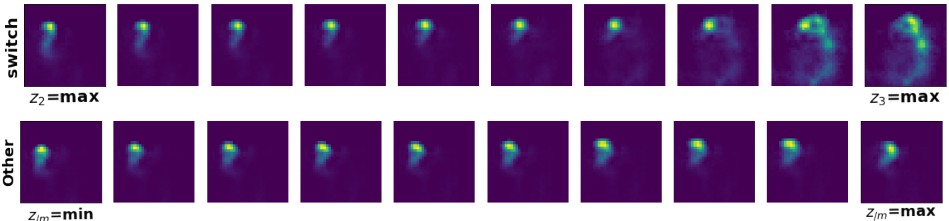

Figure 5: Traversals of the latent space learned from the DVSGesture dataset. (Top) Beginning with the right hand wave latent variable maximized and the left hand wave variable minimized, traverse along the latent space by gradually decreasing the right hand wave latent variable and increasing the left hand wave latent variable. Note the initial TS shows a small, focal area of motion in the top-left corresponding to the participant's right hand waving. (Bottom) The latent traversal along all of the non-target $z_{\setminus m}$ latent variables illustrates the relative insensitivity of the model to these features.

### 4.3 NEUROMORPHIC IMPLEMENTATION RESULTS

Figure 4 (Right) shows the latent space representations of three gesture classes mapped using the neuromorphic encoder. With just three classes, the model running on Loihi demonstrates separation between gestures and global structure, but these features are not as defined as in the conventional model. With all ten classes of gestures used to train the neuromorphic encoder, there was no clear disentanglement or obvious structure to the latent space. It is possible that the lack of separation for the more challenging 10-way task is due to the low-precision integers used for synaptic weights and membrane potentials of spiking neurons on the neuromorphic chip, thus adding variability to the embedded positions. In future work we will test our algorithm on more precise hardware and adopt powerful on-chip learning algorithms such as SOEL (Stewart et al., 2020b).

## 5 CONCLUSIONS

We presented a novel algorithm to process DVS sensor data and show that it is capable of learning highly separable, salient features of real-world gestures. The Hybrid Guided-VAE contains an encoder model that learns to represent extremely sparse, high-dimensional visual data captured at the sensor in a small number of latent dimensions. The encoder is jointly trained by two classifiers such that the latent space disentangles and accurately represents target features. The algorithm represents a significant step towards self-supervised gesture recognition by measuring the similarity of gesture data in real-time with a learned, perceptually-relevant metric. In addition, we demonstrated a first-of-its-kind implementation of the algorithm on neuromorphic hardware. Due to the sparse nature of event-based data and processing, the SNN encoding implementation offers significant benefits for applications at the edge, including extremely low power usage and the ability to learn on-device to avoid intrusive remote data aggregation. This could enable flexible gesture recognition capabilities to be embedded into home electronics or mobile devices, where computing power is limited and privacy is paramount. While the model performance currently decreases running on the neuromorphic device compared to the conventional GPU-based implementation, improvements to the neuromorphic hardware or the adoption of more powerful learning algorithms, e.g. (Stewart et al., 2020b), could alleviate these limitations. These techniques may owe some measure of their success to the computational principles they share with human perceptual systems and we expect that this approach will open new possibilities for interaction between humans and intelligent machines.

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

## A  APPENDIX

### A.1  ABLATION STUDY

We present the results of an ablation study of our Hybrid Guided-VAE method to demonstrate why we used this method for latent space disentanglement. We compare the Hybrid Guided-VAE in our work to a hybrid VAE with the guided part ablated, as well as a Guided-VAE that does not use an SNN encoder and instead use a CNN encoder, and an ordinary CNN VAE. Comparing the clustering of the hybrid VAEs and the CNN VAEs in Figure 6, both the CNN and hybrid VAEs are able to disentangle and cluster the latent space when using the guided method, with our hybrid method in Figure 6b showing less overlap between clusters and therefore better disentanglement. For classification from the latent space representation of the data, Table 2 shows that the CNN and hybrid VAEs both achieve high accuracy on both datasets, with the hybrid VAEs achieving higher performance. Therefore, our hybrid guided VAE approach is more suitable for latent space disentanglement with data taken from event-based sensors. The disentangled latent space shows the hybrid VAEs learn a measure of similarity in event-based data, such as gesture similarity, which could be used in future work for self-supervised learning from unlabeled data streamed to neuromorphic hardware using an approach similar to (Noroozi et al., 2018).

Table 2: Classification from latent space

| Algorithm | Dataset | Train | Test |
|---|---|---|---|
| Hybrid Guided VAE | DVSGesture | **97.6%** | **86.8%** |
| | NMNIST | **99.6%** | **98.2%** |
| CNN Guided VAE | DVSGesture | 86.7% | 82.3% |
| | NMNIST | 97.2% | 96.8% |
| Hybrid VAE | DVSGesture | 99.7% | 38.3% |
| | NMNIST | 96.8% | 92.4% |
| CNN VAE | DVSGesture | 97.5% | 28.4% |
| | NMNIST | 96.8% | 91.5% |

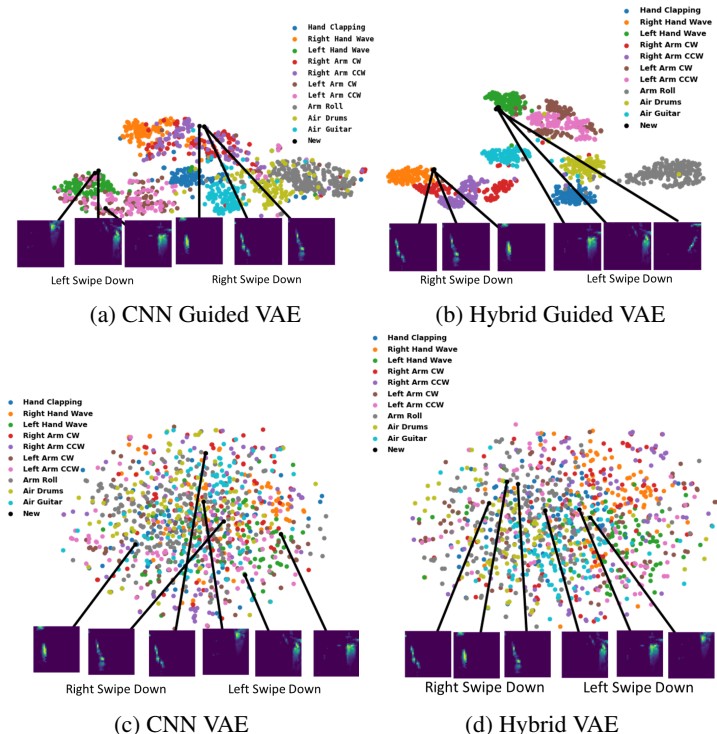

Figure 6: TSNE plots of the DVSGesture latent space. The images below each figure show novel gestures and where they are placed in the latent space by the different models.

## A.2 SNN DYNAMICS CORRESPOND TO THE DISCRETIZATION OF A LIF NEURON MODEL

We use the LIF model with current-based synaptic dynamics and a relative refractory period defined in Kaiser et al. (2020). The dynamics of the membrane potential $U_i$ of a neuron $i$ is determined by the following differential equations:

$$
\begin{aligned}
U_i(t) =& V_i(t) - U_{th}R_i(t) + b_i, \\
\tau_{mem}\frac{\mathrm{d}}{\mathrm{d}t}V_i(t) =& -V_i(t) + I_i(t), \\
\tau_{ref}\frac{\mathrm{d}}{\mathrm{d}t}R_i(t) =& -R_i(t) + S_i(t), \\
S_i(t) =& \delta(t_i^f - t_i)
\end{aligned}
\tag{7}
$$

with $S_i(t)$ representing the spike train of neuron $i$ spiking at times $t_i^f$, where $\delta$ is the Dirac delta. The constant $b_i$ represents the intrinsic excitability of the neuron. The two states $U$ and $V$ is interpreted as a special case of a two-compartment model, with one dendritic ($V$) compartment and one somatic ($U$) compartment. The reset mechanism is captured with the dynamics of $R_i$. The factors $\tau_{mem}$ and $\tau_{ref}$ are time constants of the membrane and reset dynamics, respectively. $I_i$ denotes the total

synaptic current of neuron $i$, expressed as:

$$\tau_{syn}\frac{\mathrm{d}}{\mathrm{d}t}I_i(t) = -I_i(t) + \sum_{j \in \mathrm{pre}} W_{ij}S_{in,j}(t), \tag{8}$$

where $W_{ij}$ is the synaptic weights between pre-synaptic neuron $j$ and post-synaptic neuron $i$. Because $V_i$ and $I_i$ are linear with respect to the weights $W_{ij}$, the dynamics of $V_i$ can be rewritten as:

$$V_i(t) = \sum_{j \in \mathrm{pre}} W_{ij}P_j(t),$$
$$\tau_{mem}\frac{\mathrm{d}}{\mathrm{d}t}P_j(t) = -P_j(t) + Q_j(t), \tag{9}$$
$$\tau_{syn}\frac{\mathrm{d}}{\mathrm{d}t}Q_j(t) = -Q_j(t) + S_{in,j}(t).$$

The states $P$ and $Q$ describe the traces of the membrane and the current-based synapse, respectively. For each incoming spike, each trace undergoes a jump of height 1 and otherwise decays exponentially with a time constant $\tau_{\mathrm{mem}}$ (for $P$) and $\tau_{\mathrm{syn}}$ (for $Q$). Weighting the trace $P_j$ with the synaptic weight $W_{ij}$ results in the Post-Synaptic Potentials (PSPs) of neuron $i$ caused by input neuron $j$. The equations (9) define a spike response model Gerstner et al. (2014) with linear filters.

In a digital system, the continuous dynamics above are simulated in discrete time. Therefore the equations above are discretized by integration over one time step $\Delta t$ Kaiser et al. (2020), leading to equations (7) in the main article. Discretized equations for $R$ are:

$$R_i^{t+\Delta t} = \gamma R_i^t + (1 - \gamma)S_i(t), \tag{10}$$

with $\gamma = \exp(\frac{\tau_{ref}}{\Delta t})$.

## A.3 TIME SURFACES WITH EXPONENTIAL KERNELS ARE PRE-SYNAPTIC TRACES $Q$

In continuous time, the Time Surface (TS) is defined as:

$$(\epsilon * S_{in})(t) = \int_0^t \epsilon(t - s)S_{in}(s)\mathrm{d}s \text{ with } \epsilon(s) = e^{\frac{-s}{\tau}}. \tag{11}$$

We demonstrate here that the solution of $Q$ is equal to $(\epsilon * S_{in,j})(t)$ when $\tau = \tau_{syn}$.

The differential equation governing $Q(t)$ is linear and can be integrated in a straightforward manner:

$$\tau_{syn}\frac{\mathrm{d}}{\mathrm{d}t}Q(t) = -Q(t) + S_{in}(t),$$
$$Q(t) = \int_0^t \exp(-\frac{t - s}{\tau_{syn}})S_{in}(s)\mathrm{d}s \tag{12}$$
$$= (\epsilon * S_{in})(t)$$

where we have assumed $Q(0) = 0$. The last line is true if $\tau = \tau_{syn}$.

## A.4 GUIDING ON OTHER FACTORS OF VARIATION: LIGHTING

A key feature of the guided VAE is to incorporate alternative features not directly related to the gesture class to disentangle the factors of variation in the data. To demonstrate this, in a separate experiment, we trained on the lighting condition provided in the DVSGesture dataset instead of the gesture class. The T-SNE projection of the $z_m$ latent space is shown in Figure 7. The model clusters the lighting conditions with some overlap between LED lighting and the other lighting conditions. This is likely due to the fact that the lighting conditions under which the gestures are performed are combinations of the labeled lighting conditions, with the label given to the most prominent lighting condition used Amir et al. (2017).

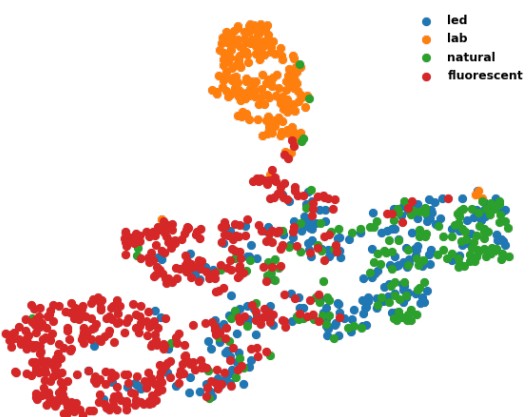

Figure 7: A T-SNE plot of the guided $z_m$ latent space using lighting condition labels.

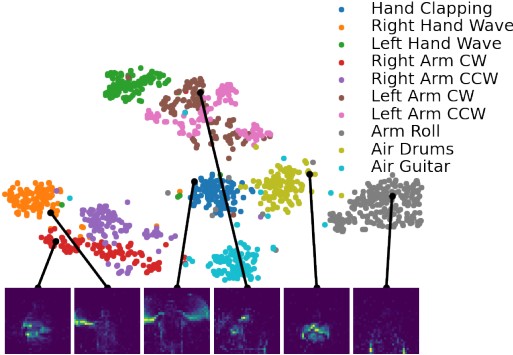

Figure 8: A T-SNE plot of the $z_m$ portion of the latent space of the encoded DVSGesture dataset . Six unlabeled gestures (labeled "other" in the dataset) are shown below as time-surfaces connected to the respective embedded positions in the latent space. Each of the unclassified gestures are embedded near gesture classes which exhibit similar motion.

## A.5 Labeling Unlabeled Gestures

Figure 8 shows a T-SNE plot of the guided portion of the latent space trained on the DVSGesture dataset. The latent space is clustered by the gesture that the excitation classifier targeted for each of the $z_m$, with some overlap between very similar gestures such as left arm clockwise versus left arm counterclockwise. The bottom of the figure shows samples from the "other" class in the Dvs Gesture dataset in relation to the T-SNE projection of the guided latent space. The embedded positions of these gestures show which gesture classes they are most similar to. For example, gestures with certain salient features, such as a salient left hand, are placed with the cluster of the gesture class that has the same salient feature, such as the left hand wave class. Thus, the latent states obtained for these gestures can provide the basis for pseudolabeling of new gestures by measuring the similarity between unclassified samples and existing gesture classes in the latent space.

## A.6 Hybrid Guided VAE Classifier Architectures

Table 3: Hybrid Guided-VAE excitation classifier architecture

| Layer | Kernel | Output | Layer Type |
|-------|--------|--------|------------|
| input |  | 10 | $z_t$ |
| 2 | - | 100 | Linear layers |
| 3 | - | 100 |  |
| output | - | 10 | Feature (One Hot Target) |

Table 4: Hybrid Guided-VAE inhibition classifier architecture

| Layer | Kernel | Output | Layer Type |
|---|---|---|---|
| input | | 90 | $z_{rst}$ |
| 2 | - | 100 | Linear layers |
| 3 | - | 100 | |
| output 1 | - | 10 | Feature (One Hot Target) |
| output 2 | - | 10 | Feature (No Hot Target) |

## A.7 SYSTEM SPECIFICATIONS FOR MEASUREMENT

The training of the hybrid Guided-VAE model and the latent space classifier were performed with Arch Linux 5.6.10, and PyTorch 1.6.0. The machine consists of AMD Ryzen Threadripper CPUs with 64GB RAM and Nvidia GeForce RTX 1080Ti GPUs. For the neuromorphic hardware implementation, an Intel Nahuku 32 board consisting of 32 Loihi chips running on the Intel Neuromorphic Research Community (INRC) cloud were used with Nx SDK 1.0. The machine consists of an Intel Xeon E5-2650 CPU with 4GB RAM.

