# OpenReview forum: "Encoding Event-Based Gesture Data With a Hybrid SNN Guided Variational Auto-encoder"
_ICLR.cc/2022/Conference — ICLR 2022 Submitted_

### Official Review · Reviewer_t5op · 2021-10-31

**Correctness:** 2
**Technical Novelty And Significance:** 3
**Empirical Novelty And Significance:** 2
**Recommendation:** 3
**Confidence:** 3

**Details Of Ethics Concerns:**

No comment

**Main Review:**

+ The work focuses on novel sensors that smartly comprises visual data, triggering events only when changes occur in the image for every location. Also, event streams have very high temporal resolution and HDR.
+ The solution explicitly takes advantage of the temporal resolution, employing a SNN that uses a variational autoencoder for the learning.
+ Authors present the results evaluating on a widely used dataset for DVS data, showing how their solution achieves correct classification for different gestures, and also showing how the clustering using the autoencoder clearly separates different sub-gestures or parts, that could potentially help with generalization as well

- The motivation for the use of event-based sensors is sometimes confusing. Specifically, authors say an event-based recognition system helps with the problem of requiring rigid gestures. However, the paper does not demonstrate how and why. Some experiments are needed to actually prove that the system presented "recognizes natural gestures despite their wide ranging variability"
- Authors do not compare with other works in the state of the art. There are multiple solutions already for gesture recognition, and specially using the DVS gesture dataset.
- There is no clear section in the experimental section that shows results with a classification metric
- Authors also mention several times their implementation of the solution in Loihi. From the paper right now, it is not clear if they implemented the whole solution (it seems it was implemented partially) and there is not data about any hardware implementation with no details about the resources, the performance, the energy consumption, etc. In fact, I would remove this part from the paper since it makes the objectives of the work a bit more diffuse

**Summary Of The Paper:**

The papers presents a method for gesture recognition using event-based sensors. Also, instead of using conventional machine learning techniques based on images, authors present a solution that employs a guided variational autoencoder that is plugged into a SNN network, making a the whole system a biologically-inspired solution.
Authors show the efficacy of their method on a dataset with event streams that is publicly available.
Finally, authors also mention they partly implemented the solution on the Intel neuromorphic chip Loihi

**Summary Of The Review:**

My recommendation is that the paper should not be published in its present form. Although the solution is novel and interesting and addresses a problem that is relevant to the community, the lack of comparison to other works in the state of the art, makes the work hard to assess (the comparison to only 1 work in the appendix is not enough either)
Secondly, the experimental section is not complete, and needs to be improved adding evaluation metrics and a proper discussion about them
Finally, the section about the implementation in the neuromorphic chip seems superficial and might need to be left out to focus the paper in the solution/method itself.

---

### Official Review · Reviewer_6Az2 · 2021-10-31

**Correctness:** 3
**Technical Novelty And Significance:** 2
**Empirical Novelty And Significance:** 3
**Recommendation:** 6
**Confidence:** 4

**Main Review:**

### Strengths
1. The proposed approach is technically sound. It is especially good that the authors show the results after implementing their network on neuromorphic hardware.

2. The experimental results show that VAE architecture learns almost perfectly separated clusters for the target gesture classes, and classification accuracy of 97% on neuromorphic MNIST dataset and 87% on the IBM DVSGesture dataset.

### Weaknesses
The full extent of the benefits offered by the different components of the proposed approach was unclear from my reading of the paper.
1. Why is the VAE architecture necessary for gesture detection? Since the authors already employ separate classifiers to disentangle the latent features, what additional benefit is provided by the decoder component for feature learning? This would have been best illustrated with an ablation study by removing the decoder from the network architecture.

2. Further, performing an ablation study consisting of only one classifier, and therefore no disentanglement of the latent features, would have clearly justified the need for disentanglement for the purpose of gesture recognition in the first place.

3. While it is appreciable that the proposed network reads event data at the resolution of 1 ms, how useful is it to read the data at such a high resolution? Have the authors performed any experiments of sampling the data at lower temporal resolutions and observing the corresponding detection performance? My understanding is that processing the input data at higher resolutions is more expensive. Moreover, for such high resolutions as 1 ms, the network may be flooded with redundant data (consisting of almost identical event data in a majority of the adjacent frames) since the rate of change of gestures is presumably much lower, hence most of the network's processing ends up being wasteful.

**Summary Of The Paper:**

The authors present a method to detect mid-air gestures by transforming event-based data captured by a Dynamic Vision Sensor (DVS) into meaningful latent representations. The proposed method consists of a variational autoencoder (VAE) built using spiking neural networks (SNNs) to encode the event data at a high temporal resolution. The VAE is coupled with a pair of classifiers to disentangle the latent features into event-aware and event-agnostic features, and detect the gestures based on the event-aware features. Experimental results on two datasets show that the proposed approach outperforms a convolutional neural network (CNN) based VAE baseline and is able to cluster the features for the different gesture classes in the latent space.

**Summary Of The Review:**

Overall, the proposed approach provides excellent results for gesture recognition on two benchmark datasets and goes the extra step of implementing the neural network on neuromorphic hardware. However, I have some questions regarding the benefits of the individual components of the neural network and concerns regarding the proportion of wasteful computation done by the network, as detailed under "weaknesses". I invite the authors to address these comments and believe that the paper would subsequently make a strong contender for acceptance.

---

### Official Review · Reviewer_LZrg · 2021-10-31

**Correctness:** 3
**Technical Novelty And Significance:** 2
**Empirical Novelty And Significance:** 3
**Recommendation:** 3
**Confidence:** 3

**Main Review:**

I believe this work brings some interesting concepts and several good ideas that could be of interest for practitioners. However I think its final outcome only goes halfway. I feel the architecture proposed is not really leveraged to prove an actual benefit over existing approaches. Experimental setup is somehow limited to prove the input is properly captured and reconstructed but does not show other benefits beyond that neither is evaluated against current SOA approaches.

As a practitioner in the HAR domain I’d rather see a less ambitious paper, but with a more complete story. I believe authors try to combine multiple relatively novel bits (DVS, SNN, VAE) but at the end it is not clear what each one brings to the table. I understand these technologies have been around for a while and the idea may be that all of them sum up for the work’s contribution. But for gesture detection/modelling, for instance, a work showing actual benefit of SNN+VAE over more traditional LSTM+CNN , with an adequate experimental setup and evaluation, would make a proper contribution.

As another suggestion, I think the manuscript’s current title and abstract are a bit misleading toward the gesture recognition/HAR domain. Honestly I do not think authors should change the domain, but the focus of the experimentation. In my opinion, having NMNIST for this paper, given it’s not gesture related and it’s limited complexity, does not contribute much. I’d rather see a more gesture/action related experimental setup.

Overall I believe the work would truly benefit from a more comprehensive evaluation, adding to the experimental setup diverse datasets, including comparatives and -as author already note- performing ablation studies to evaluate the  contribution of each component.

By the way I looked for the ablation experiments in the Appendix and was not able to find them. Probably I missed them, just mentioning it so authors can verify they are attaching the correct appendix. In any case, if you could add these experiments to a document/pdf that’d be great, make them easier to spot.

The overall writing quality of the paper is very good. It’s well written, well structured, easy to read and I consider the figures/tables included illustrate the proposal correctly.


**Summary Of The Paper:**

Authors propose a gesture modelling approach based on two fairly novel components, the use of a DVS camera and a SNN working in conjunction with a VAE. Using the DVS cameras as input in the space/time domain, this is recurrently processed by the SNN and its latent representation modelled by the VAE. Authors show examples of this representation using two DVS captured datasets. They show the feasibility of this approach for time based events scenarios.


**Summary Of The Review:**

The paper presents several interesting ideas (DVS+SNN+VAE) in the shape of the proposed architecture. The novelty component I believe is justified since I’m not familiarized with similar approaches in the gesture modeling domain. However, and this is the reason for my recommendation, the evaluation of the proposal somehow only relies on the capacity of the system for modelling the input. It’s not analyzed the contribution of each of the pieces proposed and it’s not measured its performance against other approaches. The experimental setup is what I think diminishes significantly the relevance of the work for the conference audience.

---

### Official Review · Reviewer_K3XW · 2021-11-03

**Correctness:** 1
**Technical Novelty And Significance:** 2
**Empirical Novelty And Significance:** 1
**Recommendation:** 1
**Confidence:** 4

**Main Review:**

Strengths:
1. The proposed SNN encoder method is interesting. T-SNE plots show that it is successful in disentangling the gesture classes.
2. The paper covers the literature well.
3. The hardware implementation demonstrates the low-power realization capability.

Weaknesses:
1. First of all, the motivation behind a neuromorphic system to address the rigid gesture limitation is not clear. In fact, it can also be possible to address the rigid gesture limitation with traditional sensors. Is the very high temporal resolution of DVS really necessary to successfully handle natural gestures? This motivation is not convincing. If so, authors should be able to support their claims by some experimental comparison or by a reference to a prior work.
2. This paper does not present experimental evaluation to justify the claims, which is a major drawback. It only reports accuracy on the training and test set of the DVSGesture dataset, and presents some T-SNE based analysis. Quantitative comparisons against some competing methods are required to be able to justify the effectiveness of the proposed approach.

Minor issues:
Both arxiv and journal versions of the same paper are cited: Gallego et al. 2020 & Gallego et al. 2019. Is there a reason for that?



**Summary Of The Paper:**

This paper proposes a hybrid spiking neural network (SNN) guided variational auto-encoder (VAE) in order to overcome the limitation of rigid and dramatic gestures that are required in commercial mid-air gesture recognition systems. The motivation is that this limitation causes unnatural movements and thus fatigue. Authors claim that the proposed neuromorphic system overcomes this limitation by means of event-based data of high temporal resolution, and that it enables self-supervised learning of natural mid-air gestures for on-device learning advantage. For this purpose, a VAE latent space representation is employed, which is shown to be effective in clustering and pseudo-labeling tasks. Authors obtain 87% classification accuracy on the DVSGesture dataset, and claim that their method can learn highly separable and salient features. Their work involves a neuromorphic hardware implementation as well to demonstrate the real-time potential.

**Summary Of The Review:**

Although the method and the proof-of-concept implementation is interesting and the analyses including the latent space are informative, the motivation of this work is not clear to me and the paper lacks sufficient evaluation to support the claims. I explain these weaknesses above. This paper looks like an unfinished study. Therefore my decision is rejection of the paper.

---

### Decision · Program_Chairs · 2022-01-20

**Decision:**

Reject

**Comment:**

This paper received 3 rejections and 1 marginal accept. Reviewers were unanimous in that empirical evaluation is lacking. No rebuttal was submitted and I have no reason to overturn the reviewers' decisions. I recommendation this paper be rejected.